# The Exploitation of Microbial Antagonists against Postharvest Plant Pathogens

**DOI:** 10.3390/microorganisms11041044

**Published:** 2023-04-16

**Authors:** Lamenew Fenta, Habtamu Mekonnen, Negash Kabtimer

**Affiliations:** 1Department of Biology, Debre Markos University, Debre Markos P.O. Box 269, Ethiopia; 2Department of Biology, Bahir Dar University, Bahir Dar P.O. Box 79, Ethiopia

**Keywords:** disease development, microbial biocontrol, postharvest diseases, postharvest loss

## Abstract

Postharvest disease management is vital to increase the quality and productivity of crops. As part of crop disease protection, people used different agrochemicals and agricultural practices to manage postharvest diseases. However, the widespread use of agrochemicals in pest and disease control has detrimental effects on consumer health, the environment, and fruit quality. To date, different approaches are being used to manage postharvest diseases. The use of microorganisms to control postharvest disease is becoming an eco-friendly and environmentally sounds approach. There are many known and reported biocontrol agents, including bacteria, fungi, and actinomycetes. Nevertheless, despite the abundance of publications on biocontrol agents, the use of biocontrol in sustainable agriculture requires substantial research, effective adoption, and comprehension of the interactions between plants, pathogens, and the environment. To accomplish this, this review made an effort to locate and summarize earlier publications on the function of microbial biocontrol agents against postharvest crop diseases. Additionally, this review aims to investigate biocontrol mechanisms, their modes of operation, potential future applications for bioagents, as well as difficulties encountered during the commercialization process.

## 1. Introduction

The human diet should include fruits and vegetables since they provide critical elements such as vitamins and minerals as well as antioxidant and anticancer compounds [1]. Increasing consumer awareness about diet and its health effects as well as their concerns about the safety of fruits and pesticide residues, toxins, and pathogens resulted in a larger intake of fruits and vegetables [2]. Infections caused by postharvest pathogens are currently the biggest worries for food production systems. They significantly shorten the shelf life of fruits and vegetables and cause significant deterioration during their postharvest processing, distribution, and storage. Postharvest fruit and vegetable diseases continue to have a large negative impact on the global economy, with losses estimated to be 20% in industrialized countries and over 50% in areas with storage and transportation constraints [3,4].

Significant losses of fruits and vegetables occur both in the field and during storage due to fungal spoilage. Fungal diseases linked to high moisture, low pH, high nutrients, and inherent resistance to decay after harvest are to blame for the high-degree loss of fruits and vegetables [5]. In addition to quality and monetary losses, fruits contaminated with fungi such as *Aspergillus*, *Alternaria*, *Fusarium*, and *Penicillium* pose a major health concern due to the mycotoxins they produce, including aflatoxins, ochratoxins, alternariol, and fumonisin [6].

In the past, the main technique for preventing fungus-driven postharvest deterioration was to apply agrochemicals either before or after harvest [7]. Nevertheless, the development of new pathogen biotypes, the rise of pathogen resistance to many fungicides, the increase in fungicide residue levels in agricultural production, the lack of effective substitutes, the negative effects on the environment, and toxicological issues relating to human health have made the utilization of synthetic fungicides in postharvest disease prevention a major source of concern for people in the agricultural sector [8]. People are compelled to look for safe and environmentally friendly alternatives in order to control postharvest infections and decay as a result of the aforementioned issues. The employment of antagonistic microbes for biological control is a novel and alluring alternative among the several methods for preventing postharvest infection and decay brought on by pathogens [9,10]. In comparison to synthetic fungicides, the application of antagonistic microbes in the management of postharvest disease has several advantages. They are inherently less harmful than chemical pesticides. Moreover, it affects only the target pest in contrast to broad-spectrum conventional pesticides [11]. Yet, scientific evidence indicates that these advantages are not always realized. Since microbial pesticides are living organisms, their main drawbacks include their extremely high specificity against the target disease and pathogen, which may require the use of multiple microbial pesticides, and their frequently variable efficacy brought on by the influences of various biotic and abiotic factors [11]. A number of fungal and bacterial biocontrol agents were identified for commercial use [5]. More information is also available on the formulation, fermentation, handling, and storage of biocontrol antagonists [12]. Therefore, the goal of this review is to give a comprehensive understanding of postharvest biocontrol systems driven by microbial antagonists, along with the mechanisms of biocontrol, usage, and ways to increase efficacy. Thus, the purpose of this chapter is to give a brief review of the utilization of microbial antagonists as postharvest biocontrol agents while summarizing data on their mechanisms of action, methods of application, and current constraints against their usage.

## 2. Postharvest Disease Development

Several fungal infections are the cause of postharvest disease and decay in vegetables and fruits. During storage and transportation, fungal-infected crops begin to show indications of illness. Several factors, including abiotic stressors, such as ripening, harvesting, and mechanical damage, frequently activate and lead to the development of postharvest diseases. To begin the disease-development process, fungal pathogens germinate and penetrate the host tissue cuticle through cuts and injuries [13]. The pathogenic fungi consume resources from the host while developing, killing the host tissues necrotrophically and starting the degradation of tissues.

The presence of high-water content in the orchard as a result of the water content of the plant products makes vegetables and fruits susceptible to pathogen attack. More importantly, the presence of wounds in the organs of plants produced during harvest and transport is also an ideal route for pathogenic fungi, particularly necrotrophic ones. Many bacteria and fungi typically enter through wounds or natural openings (such as lenticels or stomata). *Erwinia amylovora* causes fire blight in apples and pears (via hydathodes), *Puccinia graminis* causes stem rust in wheat (natural openings), *Streptomyces scabies* causes potato common scab, and *Penicillium expansum* causes blue mould rot [14,15,16,17]. However, certain fungal species are capably secreting specific enzymes that enable them to penetrate the intact cuticle, stems, and fruits by exerting mechanical pressure. Numerous fungal genera, including *Alternaria*, *Botrytis*, *Botryosphaeria*, *Colletotrichum*, *Lasiodiplodia*, *Monilinia*, and *Phomopsis*, are known to live inactively and go unnoticed by visual inspection while unripe fruits are being stored until they ripen. Fungal pathogens multiply rapidly as the fruit ripens [18]. Other harmful fungi may dwell on fruit tissue till ripening, either endophytically or hemibiotrophically. Meanwhile, as the fruits’ ability to resist disease is reduced, they become more susceptible to fungal infections [18]. Therefore, it becomes crucial to prevent disease before and after harvest to prevent crop damage in terms of both quantity and general quality.

### 2.1. Postharvest Diseases Management

Postharvest loss is the decline in a food product’s quality and quantity from harvest to consumption. Quantity losses, which relate to incidents that cause a product’s quantity to be lost, are more frequent in developing nations [19]. Quality losses, on the other hand, include those that affect the acceptability, nutrient/caloric composition, and edibility of a given product [20]. Crop losses and postharvest quality degradation are mostly brought on by microbial infection, pests, ripening processes that occur naturally, and environmental factors such as drought and poor postharvest handling [21,22]. Postharvest operations such as harvesting, handling, storing, processing, packaging, and distribution are to blame for the quantity and quality loss of crops [23].

Various techniques have been developed to control postharvest diseases, and they can be broadly divided into three categories: physical (low-temperature storage, heat treatments), chemical (pre- and postharvest chemical treatments), and biological (using natural plant products or antagonistic microorganisms). The intricate interactions between the host, pathogen, and environment must be managed through an integrated strategy that includes cultural, preharvest, harvest, and postharvest techniques. According to Adaskaveg et al. [24], postharvest disease control can either be preventative, such as through cultural practices, host resistance, exclusion (quarantines and sorting), reducing inoculum, protection through chemical, biological, or physical treatments; or curative, requiring therapeutic treatments. Synthetic fungicides have generally been the main tool for preventing postharvest infections. However, the rise of fungicide-resistant strains of pathogens, environmental contamination linked to pesticide use, and increased public concern over human health conditions have pushed the quest for alternate methods.

### 2.2. Biological Control

The term “biocontrol” refers to the use of antagonistic microbes to control diseases as well as the utilization of pathogens that are specific to a particular host to regulate weed populations [2]. The term “biocontrol” has been used more broadly to refer to the use of naturally occurring products that have been extracted or fermented from a variety of sources as well as the use of one or more additional organisms to control the negative traits of a single organism, also recognized as a natural enemy [25]. One of the most promising options to reduce pesticide use is biocontrol using microbial antagonists, either on their own or as a component of integrated pest management. Antagonism is a phenomenon in which antagonistic organisms act to inhibit or decrease the normal development, growth, and activity of phytopathogens that are present nearby. These organisms, also known as “Biological Control Agents”, are capable of eradicating insect pests and pathogens that harm horticulture crops [26]. Numerous microbes with antagonistic effects on preharvest and postharvest pathogens have been documented. These microbes do have a variety of antagonistic traits, such as the ability to produce pathogen-specific antifungal metabolites that suppress or eradicate the pathogen growing on fruit and prevent further fruit loss during storage [12]. Bacteria, yeast, and filamentous fungi are just a few of the taxonomic categories that include antagonistic microbes. Numerous authors have found various microbial species that have been artificially introduced as biocontrol agents on a variety of horticulture products throughout the last few decades [27]. The natural epiphytic antagonistic microflora that already exists on the surfaces of fruits and the exogenous introduction of the specific microbes with antagonistic activity are used to decrease postharvest damage via microbial antagonists [28].

The majority of microbial antagonists are found naturally on the surfaces of fruits and vegetables, where they appear to be endemic. Some of the microbial antagonists that were recovered in potatoes, tomatoes, citrus roots, mangos, and bananas include *Bacillus subtilis*, *Rhodotorula glutinis* Y-44, *Kloeckera apiculate*, *Lactobacillus acidophilus*, and *Trichoderma harzianum*, which are used to control plant pathogens [29,30,31,32,33]. Several of them are effective biocontrol agents for the management of postharvest diseases [34,35]. Other intimately associated or unrelated sources, such as the phyllosphere, rhizosphere, and soil, can also provide the microorganisms, in addition to the fruit surface [29,30,36].

Currently, a feasible alternative for fruit protection against phytopathogens at the postharvest stage is biocontrol, which offers protection against fungal diseases [37]. In laboratory investigations, numerous microbial antagonists of postharvest pathogens (fungi and bacteria) have been revealed [37]. A list of microbial antagonists recovered from various sources and used as biocontrol agents for postharvest plant diseases is indicated in Table 1.

### 2.3. Sources of Microbial Antagonists

Most of the antagonistic microbes emanate from the surface of the fruit, plant parts, sea, and soil [76]. Moreover, different fermentation products are also found to be the source of antagonistic bacteria and fungi. Numerous investigations have shown that a wide range of microbial antagonists are effective in preventing postharvest fungal infections [27,77]. For instance, lactic acid bacteria (LAB) recovered from different fermentation products are effective in controlling postharvest pathogens [78]. Trias Mansilla et al. [79] also isolated LAB from the surfaces of fruits and vegetables against *Xanthomonas campestris*, *Erwinia carotovora*, *Penicillium expansum*, *Monilinia laxa*, and *Botrytis cinerea*. Unique natural habitats such as Antarctic soil and marine environments also were found to be sources of effective microbial antagonists. For instance, the yeast *Leucosporidium scottii*, isolated from Antarctic soil, was an effective microbial antagonist against *P. expansum* and *B. cinerea* [80]. Similar to this, the yeast *Rhodosporidium paludigenum*, which was isolated from marine environments, was revealed to be efficient in *P. expansum* development on pear fruits [81]. Marine yeasts have higher osmotolerance levels than yeasts isolated from the fruit surface. Hence, marine yeasts are more suitable candidates for high abiotic stress environments [54]. Marine-environment bacteria showed a considerable amount of potential for biocontrol in peanut cultivation [82]. The effectiveness of halotolerant marine *Trichoderma* isolates in inducing a systemic defense response in plants against *Rhizoctonia solani* was assessed for possible biocontrol applications [83]. Most microbial antagonists are naturally present on the surfaces of fruits and vegetables, including *Bacillus subtilis* from potatoes, *Rhodotorula glutinis* Y-44 from tomatoes, *Kloeckera apiculate* from citrus roots, *Lactobacillus acidophilus* from mangoes, and *Trichoderma harzianum* from bananas [29,30,31,32,33]

## 3. Mechanisms of Microbial Antagonism

Knowledge about the mechanism of action of antagonism is a key factor for the effective prevention of phytopathogens in their hosts. Various modes of action are used by the microorganisms that have been proven for the biocontrol of phytopathogens [84,85].

### 3.1. Struggle for Nutrients and Space

The primary antagonistic strategy employed among many antagonists against phytopathogens is competition for carbon sources and space [86]. The availability of the carbon sources required for survival and growth limits the phytopathogenic fungus’ carbohydrate disposition and, hence, its capacity to attack the host. This limits the ability of microorganisms to damage fruit [87]. Competition for nutrients such as amino acids, carbohydrates, minerals, and vitamins, as well as for oxygen and space, is crucial to controlling postharvest fruit loss [88]. Numerous in vitro investigations have shown that competing microorganisms prevent the growth of phytopathogenic fungi by limiting their access to several carbon sources, primarily sucrose, fructose, and glucose [89]. According to a study by Yu and Lee [90], *Pseudomonas putida* prevented *Penicillium digitatum* spores from germinating as a result of the availability of nutrients. As discussed by Liu et al. [10], when yeast cells are in contact with a fruit surface they will inhabit surface wounds caused during harvest, through handling, and by quick growth as a consequence of depleting available nutrients. The interplays between *Pichia guilliermondii* and *B. cinerea* on apples [88]) and *Colletotrichum* spp. on peppers [91] were both shown to involve competition for sugars and nitrates. When there is a lack of nutrients, the antagonists reduce the amount of nutrients present at the location of the wound, preventing pathogens from germination, growth, and infection. Poppe et al. [92] found that the antagonist (*P. agglomerans* CPA-2) can inhibit the germination of conidia at low concentrations of nutrients, but not at greater quantities. The ability of antagonistic yeasts to communicate with their pathogen hyphae strengthens nutritional rivalry, which delays the onset of the pathogenic infection process [93]. By effectively colonizing the fruit surface and producing harmful metabolites, the naturally occurring nonpathogenic microbes of fruits can also affect nutrition and space competition [86]. The microbial antagonist load and the species of the host fruit also have an impact on how fast the wound site will be colonized because some antagonists require specific types of nutrients.

### 3.2. Siderophore

The biological control of pathogenic fungi depends heavily on iron (Fe^3+^), which is required for the proliferation and virulence of pathogens [93]). Numerous investigations revealed that siderophore-producing microorganisms are crucial for disease prevention. As biocontrol antagonists, *Trichoderma* species produce more potent siderophores that chelate iron (Fe^3+^) and inhibit the growth and proliferation of other diseases of fungi [94]. It was also investigated that *Rahnella aquatilis* with siderophore production hindered *B. cinerea* and *P. expansum* postharvest infections [95]. Moreover, *Metschnikowia pulcherrima* and *Monilinia fructicola* yeasts that produce siderophore pulcherrimin were effective for the biological control of postharvest apple pathogens *B. cinerea*, *Alternaria alternata*, and *P. expansum* [96]. Similarly, siderophore-producing yeast biocontrol agents, namely *M. pulcherrima* and *M. fructicola*, were found to effectively control *B. cinerea*, *A. alternata*, and *P. expansum* on apples [96]. Additionally, it has been demonstrated that siderophores production relates to the biocontrol capability of the bacterium *Rahnella aquatilis* against postharvest diseases (*B. cinerea* and *P. expansum*) of apples. Pathogens such as *B. cinerea*, *A. alternata*, and *P. expansum* were inhibited from growing mycelia and conidia germination as a result of *M. pulcherrima* iron depletion in the growth medium [95].

*Bacillus subtilis* produced siderophores, playing an important role in the control of *Fusarium oxysporum* [97]. Siderophore-producing *Azotobacter* sp. isolated from soil rhizosphere were found to be effective against fungal pathogens such as *Fusariurm* sp., *Alternaria* sp., *Phytophthora* sp., *Rhizoctonia* sp., *Colletotrichum* sp., and *Curvularia* sp [98]. Moreover, *Aureobasidium pullulans* L1 and L8 produced siderophores and to prevent thepostharvest fruit decay of peaches caused by *Monilinia laxa* [99].

### 3.3. Enzymes That Degrade the Cell Wall

Chitin, cellulose, proteins, and hemicellulose can all be broken down by microorganisms with their ability to produce enzymes. These organisms may also be used to combat plant diseases. According to research, pathogen cell walls can be hydrolyzed by enzymes including glucanases, chitinases, and proteases that are secreted by antagonistic microbes such as *Trichoderma* strains [100]. Geraldine et al. [101] revealed that N-β-acetylglucosaminidase and β-1,3-glucanase are components of the *Trichoderma* species against *Sclerotinia sclerotiorum* in the field. Moreover, it was reported that *Fusarium oxysporum*, *R. solani*, and *Botrytis species* are prevented from proliferating by the chitinase-producing *Serratia marcescens* [102]. *Bacillus* sp. capable of producing hydrolytic enzymes such as β-1,3-glucanase, protease, and chitinase was reported to be effective against Fusarium oxysporum f. sp. lycopersici [103]. Furthermore, cell-walldegrading chitinase from the *Trichoderma* species are effective against strains of *Sclerotium rolfsi* and *Colletotrichum* sp. [104].

### 3.4. Pathogen Suppression through Antibiotic Production

Some antagonists produce antibiotics to stop the spread of pathogens. The phenomenon known as antibiosis occurs when antagonists emit chemical substances that either prevent or eliminate prospective pathogens nearby. Some soil-borne microorganisms, such as *Bacillus*, fluorescent *Pseudomonas*, and *Trichoderma*, have been known to produce antibiotics for biocontrol abilities. Bacterial strains that produce antifungal antibiotics are efficiently utilized as postharvest biocontrol agents. Enzymes, exotoxins, and metabolites with nematicidal activity can be produced by *Bacillus* spp. [105]. *Bacillus* spp. also synthesizes antibacterial and antifungal metabolites such as bacillomycin, gramicidin surfactin, and fengycin [106]. *Burkholderia cepacia* was found to produce the pyrrolnitrin antibiotic which has been used against *B. cinerea, Penicillium expansum Penicillium digitatum*, and pathogens [107]. Similarly to this, *Pseudomonas syringae*’s syringomycin was employed to stop apple grey mold and citrus green mold [12]. It is also known that *B. subtilis* and *Pseudomonas cepacia* synthesize iturin, which prevents the growth of pathogenic fungi [108]. The control of *P. digitatum* in lemons and *P. expansum* and *B. cinerea* in apples were both achieved using pyrrolnitrin-producing *P. cepacia* [86]. *Streptomyces* spp., in addition to these useful microbes, can aid plants by producing antibiotics to combat phytopathogens [109]. It is unclear what role antibiotic-mediated antibiosis plays in particular biocontrol systems, despite the exploration of several antibiotic-producing microbial antagonists for the prevention of postharvest pathogens [110]. To manage postharvest crop diseases, increased focus is given to the utilization of microbial biocontrol agents that do not produce antibiotics. Furthermore, an antagonist that depends on antibiotic release may gradually lose its strength throughout the fruit’s late-stage storage phase, which only makes problems worse [111]. This strategy could be more widely accepted and prevent the rapid growth of pathogen resistance to these antibacterial substances [12]. Currently, the indirect mode of action through the induction of resistance in the host plant is given great attention in terms of disease control [112]. Induced resistance presents the possibility of long-term and thorough disease management by utilizing the intrinsic disease resistance of plants. Based on variations in signaling pathways and efficacy spectra, induced resistance has been divided into two types: systemic acquired resistance (SAR) and induced systemic resistance (ISR) [113].

### 3.5. Lytic Enzyme Production and Mycoparasitism

Fungal propagules either die completely as a result of mycoparasitism or have their structure destroyed and lysed [114]. The process of mycoparasitism involves close contact with the pathogen, mutual recognition between the pathogen and antagonist, the release of lytic enzymes by the antagonist, penetration of the host, active development of the antagonist inside the host, and exit [93,115]. Initial contact and recognition between the antagonist and the pathogen are mediated by various chemical compounds such as lectins; the penetration step is achieved by cell-wall-degrading enzymes (CWDEs), such as chitinases, β-1,3-glucanases, lipases, and proteinases [116]. For antagonists to function as biocontrol agents, pathogenic fungi’s cell walls must be broken down by extracellular hydrolytic enzymes such as chitosanases, chitinases, cellulases, and/or proteases, either singly or in combination [88]. The disintegration of fungal pathogens hyphae by the enzymatic activity of antagonists causes cellular deformities, thereby resulting in cytological damage, mycelial lysis, deformation, increased cell membrane permeability, and cytoplasmic content leakage [86]. The extracellular enzymatic activity of numerous microbes was implicated in their antifungal action [117]. According to Urbina et al. [118], biocontrol of *P. expansum* in apples is facilitated by extracellular exo-b-1, 3-glucanase from yeast *C. oleophila*. A purified glucanase enzyme, according to these scientists, prevented the growth of pathogen mycelia and decreased conidial germination. Mycoparasitism has been linked to alkaline serine protease, which is produced by the yeast-like fungus *A. pullulans* [119]. The degradation of pathogenic fungi’s cell wells has also been attributed to extracellular enzymes released by *Trichoderma*, including endochitinases, β-1,3-glucanases, and proteases [116].

### 3.6. Induction of Host Resistance

The use of microbial biocontrol agents on fruit surfaces has been shown to cause systemic resistance (ISR) against invasive fungal infections in several studies [120,121,122]. Building up structural barriers and eliciting a variety of biochemical and molecular defense mechanisms are two steps in the process of a host’s ability to adapt to biotic or abiotic stressors [123,124]. In this regard, *Bacillus* and *Pseudomonas* spp. take first place. The safety of cotton plants against *Meloidogyne incognita* and *Meloidogyne arenaria* is a result of *Bacillus subtilis* stimulation of the ISR. *Pseudomonas putida* and *S. marcescens* stimulated the ISR, which prevented *F. oxysporum* f.sp. cucumerinum from causing cucumber fusarium wilt. Additionally, ISR against *P. syringae* pv. lachrymans has been established by *S. marcescens* and *Bacillus pumilus* [125]. Additionally, preharvest exposure to the antagonistic yeast *R*. *paludigenum* on mandarins increased the synthesis of enzymes such as b-1, 3-glucanase, phenylalanine ammonia-lyase (PAL), peroxidase (POD), and polyphenol oxidase (PPO) [88,126]. It has not yet been completely proven that the induction of host defense and the inhibition of pathogenic growth are related. The molecular profile of the genes responsible for antagonistic microorganism, host, and pathogen interactions for the induction of host resistance remains a future assignment to be studied by advanced molecular studies. The aspartic protease P6281 secreted by the fungus *Trichoderma harzianum* plays an important role in mycoparasitism on phytopathogenic fungi [127]. Many *Trichoderma* spp., including *Trichoderma virens*, *Trichoderma atroviride*, and *Trichoderma harzianum*, can induce both localized and systemic resistance in a range of plants to a variety of plant pathogens [128].

### 3.7. Production of Volatile Compounds

Volatile compounds (VOCs) are one of the several antifungal metabolites produced by microbial antagonists, which are crucial in preventing the growth of fungal pathogens [129]. Fruit postharvest diseases are controlled by VOCs produced by fungi, yeast, and bacteria [86,130,131]. The ability of VOCs produced by *Bacillus* spp. to inhibit the growth of fungal pathogens is well documented. VOCs made by *B. thuringiensis* and *B. pumilus* decreased anthracnose infections in mangos by about 88.5% [131]. Similar to this, *B. subtilis* VOCs during in vitro tests decreased *P. digitatum* growth by 30–70% [132]. VOCs synthesized by *B. amyloliquefaciens* and *B. subtilis* were tested for antifungal efficacy against the citrus *Penicillium* infection by Arrebola et al. [106]. Many morphological abnormalities, including altered cell vacuolation, altered membrane permeability, and swelling in the hyphae, were discovered through electron microscopy of the pathogen hyphae subjected to the volatile compounds. These abnormalities led to poor conidia germination and appressorial development [133]. Biocontrol ability and volatile organic compound production as a putative mode of action of yeast strains isolated from organic grapes and rye grains were reported by Choińska et al. [134]. Ethyl esters of medium-chain fatty acids, phenylethyl alcohol, and its acetate ester were among the VOCs emitted by yeasts in the presence of the target plant pathogens such as *Mucor* spp., *Penicillium chrysogenum*, *Penicillium expansum*, *Aspergillus flavus*, *Fusarium cereals*, *Fusarium poae*, as well as *Botrytis cinerea*.

### 3.8. Biofilm Formation and Quorum Sensing

Antagonistic bacteria need to possess certain traits that make it easier for them to cling to fruit surfaces and to colonize and multiply inside the host. Most of the time, these traits are related to the development of biofilms. The biofilms that are created serve as barriers between the phytopathogen and the host lesion surface. The newly generated microcolonies are capable of maintaining a sort of communication via quorum sensing, using a variety of chemical signals to monitor their surroundings, changing the expression of their genes, and gaining an advantage over rivals [135]. Unfortunately, little is understood about the primary processes and mechanisms underlying the development of biofilms [136]. The *B. subtilis* strain ATCC6051 was found to be able to create biofilm in the roots of Arabidopsis plants, protecting them from infection by the bacterium *Pseudomonas syringae* [137]. *Paenibacillus polymyxa* colonizes plant roots, creating what Haggag and Timmusk [138] showed to be biofilm-like structures that shield the roots from diseases brought on by phytopathogens.

## 4. Application Methods of Biocontrol Agents

The most crucial factors in postharvest pathogen control are timing the application correctly and choosing the right method. There are two types of application approaches preharvest and postharvest which are commonly practiced combatting post-harvest crop pathogens.

### 4.1. Preharvest Application

To combat postharvest infections, microbial antagonists are either artificially introduced or those that already exist on the produce and can be managed and fostered [4]. There is strong evidence that various pathogen-contaminate fruits and vegetables are in the field and that these infestations have a significant impact on the decay of the commodities throughout transportation or storage [139]. Preharvest administration of microbial antagonists can increase biocontrol efficacy because preharvest application gives plenty of interaction time between the antagonist and the pathogen [140]. In the study conducted by Teixidó et al. [141], the incidence of blue mold produced by *P. expamsum* on damaged apples during cold storage was reduced by 50% after inoculation with the antagonistic yeast *Candida sake* CPA-1 48 h before harvesting in the field. Similar to this, Cañamás et al. [142] found that the use of *P. agglomerans* in the preharvest stage effectively safeguarded the *P. digitatum* pathogen during the storage of oranges. Furthermore, the use of *Epicoccum nigrum* in field settings proved effective in preventing the development of brown rot in postharvest peach fruits [143]. Treatments of antagonistic yeasts such as *Rhodotorula glutinis, Cryptococcus laurentii, Tricho sporonpullulans* [144], *Trichoderma harzianum* [145], and *Epicoccum nigrum* [143] before harvest were discovered to be even more effective than synthetic fungicides at controlling strawberry blight after harvest. *Aureobasidium pullulans* were used before harvest in a different study, and they significantly reduced storage rots in strawberries [146], apples [147], grapes, and cherries [148]. It has been reported that reducing storage rots in pears can be achieved by applying yeast strains *Cryptococcus laurentii* and *Candida oleophila* in the field [149]. Similar to this, Cañamás et al. [142] showed that applying varying concentrations of *Pantoea agglomerans* before harvest was successful in preventing *Penicillium digitatum* during orange storage.

Another approach to enhance the management of postharvest infections is the use of a consortium of antagonists prior to harvest. For instance, the combined preharvest administration of the yeast *C. sake* and the antagonist *Pseudomonas syringae* to apples and pears improved their total biocontrol efficiency against *P*. *expansum* (Teixidó et al. [141]. However, to increase postharvest biocontrol through the use of antagonists in the field, these agents must be able to endure environmental difficulties such as nutrient deficiency, extreme heat, water stress, ultraviolet radiation, and climatic changes [110]. The antagonist’s tolerance to environmental stresses may be increased through genetic modifications and physiological enhancements.

### 4.2. Postharvest Application

Postharvest application of antagonistic microorganisms is a common approach that is practiced for controlling the postharvest disease of crops. Antagonists are either sprayed directly onto the surfaces of the vegetables and fruits or administered by dipping when they are utilized during postharvest [4]. Numerous studies have shown that the postharvest utilization of microbial antagonists is more effective than the preharvest approach for controlling diseases in fruits and vegetables, including apples [150], citruses [151,152], bananas [153], mangos [154], tomatoes [155], cabbages [156], and peaches [157]. Globally, both preharvest and postharvest administrations of antagonists are common, but postharvest applications of potential antagonists most frequently result in a significant decrease in fungal spoilage.

## 5. Biocontrol: Status, Challenges, and Prospects

Microbial biocontrol is, nowadays, being given critical attention for the control of various crop infections. Currently, a number of biocontrol microbes such as bacteria and fungi (yeast and *Trichoderma*) are isolated and tested to be effective against many plants’ pathogenic diseases. Unfortunately, the majority of biological agents work effectively in a lab setting, but fall short when applied to field conditions. The physiological and ecological constraints on the potency of biocontrol agents most likely account for this. Genetic engineering and other molecular methods present a novel opportunity for enhancing the choice and evaluation of biocontrol agents as a solution to this issue. Several techniques can help boost a bioagent’s effectiveness, such as protoplasm fusion with polyethylene glycol or mutation. Additionally, it is urgently necessary to generate bioagents in large quantities, comprehend how they work, and assess the environmental elements that encourage the rapid development of biocontrol agents. However, parallel to the improvement of the already-identified microbial biocontrol, further exploitation of potential antagonists with multiple beneficial traits should be done [136].

The promise of biocontrol has not yet been fully realized, despite being crucial to the management of crop diseases today. This is because research in this area is still confined to the lab and very little emphasis has been placed on the commercial formulations of biocontrol agents. Thus, it is necessary to assess the efficacies of biocontrol agents in pilot, semicommercial, and large-scale commercial studies under various packing conditions [5]. It is possible to prepare both dry and liquid formulations, increasing both biocontrol effectiveness and shelf life [158].

For the prevention of postharvest disease, several antagonistic microorganisms have been found; however, only a small number have been developed and made available for purchase. These products are authorized to be used against a variety of horticulture crop postharvest fungal diseases. For instance, a formulation based on the *M. fructicola* yeast strain Shemer has been used to control *Rhizopus*, *Aspergillus*, *Botrytis*, and *Penicillium*-related fungal infections [159].

Recent advances in DNA and proteomics-based technologies, combined with bioinformatics, have opened up new possibilities for the study of postharvest biocontrol systems. These developments have made it possible to better understand the molecular relationships between microbial biocontrol agents, pathogens, and hosts [160]. Moreover, improvements and advancements in numerous “omics” technologies, such as metagenomics, transcriptomics, and proteomics, may be better utilized for the in-depth elucidation of the disease-inhibitory processes of biocontrol.

## 6. Concluding Remarks

Eco-friendly technologies are presently gaining a lot of attention as a consequence of the development of pathogens that are fungicide resistant and the presence of dangerous residues in vegetables and fruits. The utilization of biological and integrative techniques for postharvest disease control has advanced significantly during the last few decades. Nonetheless, one of the key issues for regulatory bodies and consumers has been and will continue to be pesticide residues in fresh fruits and vegetables. Hence, a top research focus continues to be the development of alternative management measures to reduce the pre- and postharvest utilization of synthetic chemical fungicides. This review paper provides a short overview of the potential of biocontrol agents as a possible alternative to synthetic pesticides, along with information on their mode of action, application technique, and future implementations. Despite the presence of huge work on microbial biocontrol, there is still a gap in the effectiveness and utilization of this microbial biocontrol as many of the works are confined only to a laboratory basis. As a result of this, only a few microbial biocontrols are commercialized and get accessed by users.

Economically feasible and workable microbial biocontrol agents must be developed to fully exploit biocontrol. These include developing high-quality, cost-effective techniques of fermentation and formulation, maintaining the viability of the cell and efficacy, creating a successful marketing outlet, improving and enhancing biocontrol efficacy under commercial conditions, and gaining a basic comprehension of how biocontrol systems function and how their environment influences relationships between the biocontrol agent and host. More investigation is required into the isolation of possible microbial antagonists that exhibit a wide range of antagonistic potential across various products, their improvement, basic comprehension of postharvest biocontrol mechanisms, and their impacts on the environment. Additionally, there is still work to be done in the areas of developing cost-effective means of large-scale production and microbial antagonist formulation.

## Figures and Tables

**Table 1 microorganisms-11-01044-t001:** Microbial antagonists recovered from various sources and used as biocontrol agents for postharvest plant diseases.

Biocontrol Agents	Hosts	Phytopathogens	Diseases	References
*Pichia membranifaciens* and *Wickerhamomyces anomalus*	Apple	*Botrytis cinerea* and *Penicillium italicum*	Blue mold	Błaszczyk et al. [38]
*Pseudomonas fluorescens*	Apple	*Penicillium expansum*	Blue mold	Wallace et al. [39]
*Bacillus amyloliquefaciens*	Apple	*Penicillium expansum*	Blue mold	Calvo et al. [40]
*Rhodosporidium fluviale*	Apple	*Botrytis cinerea*	Gray mold	Sansone et al. [41]
*Aureobasidium subglaciale*	Apple	*Botrytis cinerea*, and *Penicillium expansum*	Apple rot	Zajc et al. [42]
*Bacillus amyloliquefaciens* and *Pseudomonas* sp.	Apple	*Monilinia fructigena*	Brown rot	Lahlali et al. [43]
*Serratia plymuthica*	Apple	*Botryosphaeria dothidea*	Ring rot	Sun et al. [44]
*Yamadazyma mexicana*	Avocado	*Colletotrichum gloeosporioides*	Anthracnose	González-Gutiérrez et al. [45]
*Bacillus velezensis*	Banana	*Colletotrichum musae*	Anthracnose	Damasceno et al. [46]
*Pseudomonas syringae*	Citrus	*Penicillium digitatum*	Green mold	Panebianco et al. [47]
Yeast	Citrus	*Penicillium italicum*	Blue mold	Da Cunha et al. [48]
*Saccharomyces cerevisiae*	Citrus	*Colletotrichum acutatum*	Fruit drop	Lopes et al. [49]
*Candida pyralidae* and *Pichia kluyveri*	Grapes	*Colletotrichum acutatum*	Spoilage of grapes	Mewa-Ngongang et al. [50]
*Bacillus* sp.	Grape	*Botrytis cinerea*	Gray mold	Kasfi et al. [51]
*Bacillus amyloliquefaciens*	Grapes	*Botrytis cinerea*	Gray mold	Zhou et al. [52]
*Lactobacillus plantarum*	Grapes	*Botrytis cinerea*	Gray mold	Chen et al. [53]
*Stenotrophomonas rhizophila*	Mango	*Colletotrichum gloeosporioides*	Anthracnose	Hernandez Montiel et al. [54]
*Trichoderma harzianum*	Mango	*Colletotrichum gloeosporioides*	Anthracnose	Alvindia [31]
*Trichoderma* spp.	Mango	*Sclerotium rolfsii*	Anthracnose	Bastakoti et al. [55]
*Pseudomonas synxantha*	Peach	*Monilinia fructicola*	Brown rot	Aiello et al. [56]
*Rhodotorula minuta*	Citrus	*Geotrichum citri-aurantii*	Sour rot	Ferraz et al. [57]
*Penicillium citrinum*	Mango	*Colletotrichum gloeosporioides*	Anthracnos	Sandy [58]
*Lactobacillus acidophilus*	Mango	*Colletotrichum gloeosporioides*	Anthracnos	Fenta and Kibret [32]
*Streptomyces* sp.	Mango	*Colletotrichum gloeosporioides*	Anthracnose	Zhou et al. [59]
*Bacillus amyloliquefaciens*	Mango	*Colletotrichum gloeosporioides*	Anthracnose	Liang et al. [60]
*Bacillus amyloliquefaciens, Bacillus pumilus* and *Bacillus subtilis*	Orange and lemon	*Penicillium digitatum* and *Penicillium italicum*	Green and blue mold	Hammami et al. [61]
*Clavispora lusitaniae*	Lemon	*Penicillium digitatum*, *Penicillium italicum*, and *Geotrichum citriaurantii*	Green moldBlue mold	Pereyra et al. [62]
*Bacillus velezensis*	Rice	*Aspergilus flavus*	Rice mold	Li et al. [63]
*Trichoderma harzianum*	Orange	*Penicillium digitatum*	Green mold	Ferreira et al. [64]
*Bacillus* sp.	Orange	*Penicillium digitatum*	Green mold	Tian et al. [65]
*Pseudomonas fluorescens*	Orange	*Penicillium italicum*	Blue mold	Wang et al. [66]
*Lactobacillus sucicola*,	Orange	*Penicillium digitatum*	Green mold	Ma et al. [67]
*Rhodotorula mucilaginosa*	Orange	*Penicillium digitatum*	Green mold	Ahima et al. [68]
*Bacillus subtilis*	Peanuts	*Aspergillus flavus*	Mycotoxins	Ling et al. [69]
*Pseudomonas aeruginosa*	Pepper	*Colletotrichum truncatum*	Anthracnose	Sandani et al. [70]
*Streptomyces philanthi*	Pepper	*Colletotrichum gloeosporioides*	Anthracnose	Boukaew et al. [71]
*Bacillus Subtilis*	Potato	*Fusarium oxysporum*	Dry rot	Lastochkina et al. [33]
*Bacillus* sp.	Potato	*Fusarium oxysporum*	Fusarium rot	Ntemafack et al. [72]
*Aureobasidium pullulans*	Tomato	*Aspergillus flavus*	Spoilage	Podgórska-Kryszczuk [73]
*Torulaspora indica*	Tomato	*Alternaria arborescens*	Tomato rot	Bosqueiro et al. [74]
*Lactobacillus plantarum*	Strawberry	*Botrytis cinerea*	Grey mold	Chen et al. [75]

## Data Availability

Not applicable.

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
