# Peer review of "The Exploitation of Microbial Antagonists against Postharvest Plant Pathogens"

_microorganisms, 2023, doi:10.3390/microorganisms11041044_

Round 1
Reviewer 1 Report
The review " The exploitation of microbial antagonists against post-harvest plant pathogens authored by Fenta, Mekonnen and Kabtimer, presents an literature overview mainly of the past decade of several approaches, how to deal with post-harvest pathogens of crops.
Thereby they focus on favorable, eco-friendly strategies, which avoid the use of pesticides, namely the application of biocontrol agents, i.e. microorganisms, including bacteria, fungi and actinomycetes. In detail the authors summarize work on function and mechanisms of microbial biocontrol agents, their modes of operation, as well as potential future applications and problems during commercialization.
I think the authors did a good job and present an exhaustive information of this subject which will be helpful for a specific part of the scientific community.
Nevertheless there are some points, which need to be corrected:
Page 1, abstract, line 11, "...locate and synthesize earlier publications..." I think it should be summarize instead of synthesize!
Page 2, line 1, "agricultural produce" - it should be "production"
Page 2, line 10, The statement "biocontrol leaves no toxic residues..." is more a wish than a fact. This needs to be more explored and scientific evidence should be cited in order to substantiate this view. If the authors are not successful in finding citations, they need to discuss this important issue!
Page 3, 4. Biological control, last paragraph, line 2: "...where they appear to be endemic..." please add a citation of work, which gives examples of endemic microbial antagonists on surfaces of fruits and vegetables.
Page 3, 4. Biological control, last paragraph, line 4, change "roots" into "rhizosphere" !
Page 3, 4, 5: The chapter 5 is redundant, nothing new is said compared to chapter 4. Point 4 and 5 can be united and the paragraph "Currently, a feasible alternative... shortened and rewritten.
Table 1: There is no description of the table, no explanation in the text.
Page 5: Chapter 7, line 2: The sentence "Numerous investigations have shown....." should be moved to chapter 6. It doesn't deal with a mechanism.
Page 6, Chapter 7.1., last sentence "...how fast the wound site colonizes.. " change to "how fast the wound site will be colonized..."
Page 7, chapter 7.4, last sentence: Emphasizing, that microbial biocontrol agents should preferably not produce antibiotics, marks a very good point! The authors should discuss, which strategies are appropriate or refer to the coming chapters, like 7.8. ....
Page 7, third last line: PAL, POD, PPO - no shortcuts here, write full names.
Page 10, chapter 9, last sentence: I think in naming all possible omics technologies is a bit overdone. There is an intrinsic danger, that it looks ridiculous.
Chapter 10, Concluding remarks, line 8 and 9: "...review has provide..." better use present tense here! "...the review provides...."
Chapter 10, Concluding remarks, last paragraph, line 4: "...creating a successful marketing outlet, improving, and enhancing biocontrol..." leave out the comma after improving.
Chapter 10, Concluding remarks, fourth last line: "....potential on various produce..." change produce to products ?
Organization of the manuscript:
I think a reorganization of the chapters and headings (numbering, font size....) is necessary: Chapter 2 is headed by using a bigger font size! This is confusing, since there are no chapters like 2.1... following. Chapter 3 could be headed by a subtitle 2.1. i.e. Chapter 4 is headed by Biologcal control, followed by chapter 5. "Biocontrol against.... " here again with bigger font size! That makes no sense! Use chapters and subchapters like Chapter 7.
Altogether the paper is well written and informative. The text should be improved by making the above mentioned corrections and is then ready for publication in the journal "Microorganisms".
Author Response
Dear Reviewer,
Thank you for your invaluable comments.
We, the authors of this manuscript, have thoroughly revised the manuscript to improve its readability, and all the other comments given by reviewer have been incorporated accordingly.
Page 1, abstract, line 11, "...locate and synthesize earlier publications..." I think it should be summarize instead of synthesize!
Authors’ Replies : Your viewpoint has been incorporated into the abstract (Line 11).
Page 2, line 1, "agricultural produce" - it should be "production"
Authors’ Replies : The comment has been considered and the word produce has been changed (Page 2, paragraph 1, line1).
Page 2, line 10, The statement "biocontrol leaves no toxic residues..." is more a wish than a fact. This needs to be more explored and scientific evidence should be cited in order to substantiate this view. If the authors are not successful in finding citations, they need to discuss this important issue!
Authors’ Replies : Well taken, the sentence is revised as per the comments (Page 2, paragraph 1, line 10-17).
Page 3, 4. Biological control, last paragraph, line 2: "...where they appear to be endemic..." please add a citation of work, which gives examples of endemic microbial antagonists on surfaces of fruits and vegetables.
Authors’ Replies : The reference has been taken into consideration, and the argument has been well-expressed (Page 3, paragraph 4, line 2-5).
Page 3, 4. Biological control, last paragraph, line 4, change "roots" into "rhizosphere" !
Authors’ Replies : Well taken, changed in accordance with the remark (Page 4, paragraph 1, line 1)..
Page 3, 4, 5: The chapter 5 is redundant, nothing new is said compared to chapter 4. Point 4 and 5 can be united and the paragraph "Currently, a feasible alternative... shortened and rewritten.
Authors’ Replies : Well taken, points 4 and 5 are merged and summarised (Page 4, paragraph 2, line 1).
Table 1: There is no description of the table, no explanation in the text.
Authors’ Replies : Well taken, the table is explained and cited in the document (Page 4, paragraph 2, line 4-8).
Page 5: Chapter 7, line 2: The sentence "Numerous investigations have shown....." should be moved to chapter 6. It doesn't deal with a mechanism.
Authors’ Replies : Well taken, the sentence shifted to the source of microbial antagonists (Page 5, paragraph 1, line 3-5).
Page 6, Chapter 7.1., last sentence "...how fast the wound site colonizes.. " change to "how fast the wound site will be colonized..."
Authors’ Replies : Well taken, corrected as per the comments (Page 6, paragraph 3, line 25).
Page 7, chapter 7.4, last sentence: Emphasizing, that microbial biocontrol agents should preferably not produce antibiotics, marks a very good point! The authors should discuss, which strategies are appropriate or refer to the coming chapters, like 7.8. ....
Authors’ Replies : Well taken, preferable biocontrol strategies are described as per the comments (Page 8, paragraph 1, line 1-11).
Page 7, third last line: PAL, POD, PPO - no shortcuts here, write full names.
Authors’ Replies : As per the feedback, the complete form is used for the first time in the text (Page 8, paragraph 3, line 12-13).
Page 10, chapter 9, last sentence: I think in naming all possible omics technologies is a bit overdone. There is an intrinsic danger, that it looks ridiculous.
Authors’ Replies : Well taken, the sentence is revised as suggested (Page 9, paragraph 1, line 2-6).
Chapter 10, Concluding remarks, line 8 and 9: "...review has provide..." better use present tense here! "...the review provides...."
Authors’ Replies : Well taken, corrected as per the comments (Page 11, paragraph 5, line 8-9).
Chapter 10, Concluding remarks, last paragraph, line 4: "...creating a successful marketing outlet, improving, and enhancing biocontrol..." leave out the comma after improving.
Authors’ Replies : Well taken, the comma is removed as per the comments (Page 11, paragraph 6, line 3).
Chapter 10, Concluding remarks, fourth last line: "....potential on various produce..." change produce to products ?
Authors’ Replies : Well taken, the word is corrected as per the comment. (Page 11, paragraph 6, line 8).
Organization of the manuscript:
I think a reorganization of the chapters and headings (numbering, font size....) is necessary: Chapter 2 is headed by using a bigger font size! This is confusing, since there are no chapters like 2.1... following. Chapter 3 could be headed by a subtitle 2.1. i.e. Chapter 4 is headed by Biologcal control, followed by chapter 5. "Biocontrol against.... " here again with bigger font size! That makes no sense! Use chapters and subchapters like Chapter 7.
Authors’ Replies : Well taken, the chapters are rearranged as per the comments (Part 1, 2, 3, 4, 5 and 6).

Reviewer 2 Report
This review cannot be recommended for publication without extensive revision. This review lacks many relevant details about the target of the study. Another aspect is that the article shows a somewhat unstructured presentation of the topic. My revision suggestion includes the following:
1. Introduction
- due to the mycotoxins they produce, including aflatoxins, ochratoxins, alternaria, and fumonisin [6], Change to '' due to the mycotoxins they produce, including aflatoxins, ochratoxins, alternariol, and fumonisin [6]''.
- Biocontrol leaves no toxic residues, is safe to apply, is simple to distribute, and are inexpensive to produce. This sentence is entirely not accurate, please review and correct it.
2. Postharvest disease development
- Many bacteria and fungi typically enter through wounds or natural openings (such as lenticels or stomata). Please give some examples.
3. Postharvest diseases management
- This is a rather vague statement; more specific information is needed here.
4. Biological Control
- This statement also requires further elaboration.
5. Biocontrol against postharvest pathogens of different crops
- This section is poorly written. Please explain the importance of the topic with related references.
6. Sources of microbial antagonists
- Some further explanation for this point is required.
7. Mechanisms of microbial antagonism
- Please add more explanation for most of these mechanisms such as Siderophore, Enzymes that degrade the cell wall, Induction of host resistance, and Production of volatile compounds (VOCs).
Author Response
Dear Reviewer,
Thank you for your invaluable comments.
We, the authors of this manuscript, have thoroughly revised the manuscript to improve its readability, and all the other comments given by reviewer have been incorporated accordingly.
- Introduction
due to the mycotoxins they produce, including aflatoxins, ochratoxins, alternaria, and fumonisin [6], Change to '' due to the mycotoxins they produce, including aflatoxins, ochratoxins, alternariol, and fumonisin [6]''.
Authors’ Replies: Well taken, the word is changed to alternariol (Page 1, paragraph 2, line 6).
Biocontrol leaves no toxic residues, is safe to apply, is simple to distribute, and are inexpensive to produce. This sentence is entirely not accurate, please review and correct it.
Authors’ Replies: Well taken, the sentence is revised as per the comments (Page 2, paragraph 1, line 10-17).
- Postharvest disease development
Many bacteria and fungi typically enter through wounds or natural openings (such as lenticels or stomata). Please give some examples.
Authors’ Replies: Well taken, the examples are indicated as per the comments (Page 2, paragraph 3, line 6-8).
- Postharvest diseases management
This is a rather vague statement; more specific information is needed here.
Authors’ Replies: Well taken, The statement is revised as suggested (Page 3, paragraph 2, line 6).
- Biological Control
This statement also requires further elaboration.
Authors’ Replies: Additional information about the major source of microbial antagonists is indicated (Page 3, paragraph 4, line 4).
- Biocontrol against postharvest pathogens of different crops
This section is poorly written. Please explain the importance of the topic with related references.
Authors’ Replies: Well taken, this part is revised and merged with part of biological control (Page 4, paragraph 2, line 1).
- Sources of microbial antagonists
Some further explanation for this point is required.
Authors’ Replies: Well taken, this part is updated as per the comments (Page 6, paragraph 1, line 8-11).
- Mechanisms of microbial antagonism
Please add more explanation for most of these mechanisms such as Siderophore, Enzymes that degrade the cell wall, Induction of host resistance, and Production of volatile compounds (VOCs).
Authors’ Replies: Well taken, the mechanism of microbial antagonist is updated with additional information as per the comments (Page 7, paragraph 2, line 1-6; paragraph ).

Round 2
Reviewer 1 Report
After thorough revision by the authors the review paper is now ready for publication. Two small English corrections are left and should be made: 1. Abstract line 11 - summarize instead of summarized, 2. Concluding Remarks: provides instead of provide.
Reviewer 2 Report
Thanks for considering the suggestions from the previous version. The manuscript is acceptable from my point of view. I recommend accepting it for publication.